

# Comparative efficacy of targeted therapy, chemotherapy and their combination for advanced cholangiocarcinoma: a systematic review and network meta-analysis

Zhoujun Liao[1],*, Zhuoyue Yao[2],*, Zhiqing Yang[3], Shaohua Yang[4], Wenjuan Gu[5], Huijie Wang[6] and Lingyan Deng[7]

[1] Department of General Surgery, Panzhihua College Affiliated Hospital, Panzhihua University, Panzhihua, Panzhihua, China
[2] Beijing TianTan Hospital, Beijing, China
[3] Jiangxi Dexing Hospital, Shangrao, China
[4] The First People's Hospital of Qujing City, Qujing, China
[5] Tianjin Kanghui Hospital, Tianjin, China
[6] Jincheng People's Hospital of Shanxi Province, Jincheng, China
[7] Maternal and Child Healthcare Hospital of Guangxi Zhuang Autonomous Region, Nanning, China
* These authors contributed equally to this work.

Corresponding authors
Huijie Wang,
18703565082@163.com
Lingyan Deng, 554107434@qq.com

## ABSTRACT

**Background:** Cholangiocarcinoma is a malignant tumor with a poor prognosis. Multiple randomized controlled trial (RCT) have shown conflicting benefits of different therapies. The study was to assess the effectiveness of chemotherapy (CT), targeted therapy (TT) and both of them (targeted therapy + chemotherapy; TT+CT) for advanced cholangiocarcinoma by a systematic review and network meta-analysis.

**Methods:** PubMed, EmBase, Medline, Cochrane were searched. Two reviewers independently selected published reports of RCT comparing any targeted therapy, chemotherapy and targeted therapy combined with chemotherapy *vs.* placebo. The outcomes were overall survival (OS) and progression-free survival (PFS) on the hazard ratio-scale (HR) and mean differences-scale (MD).

**Results:** We included 13 RCT involving 1,914 patients. We revealed a differential reporting of outcomes. All three treatments significantly reduced the HR in OS and PFS when compared with the placebo. HR and MD values of OS and PFS in TT+CT group were significantly better than those in the other two groups. Only targeted therapy can significantly improve PFS in patients.

**Conclusions:** 1) All treatments significantly reduced the hazard ratio in OS and PFS when compared with the placebo; 2) HR and MD values of OS and PFS in TT+CT group were significantly better; 3) Only targeted therapy alone significantly increased the PFS of patients, thereby improving the quality of life of patients.

## INTRODUCTION

Cholangiocarcinoma is a highly malignant tumor of epithelial origin (*El-Diwany, Pawlik & Ejaz, 2019*), its global incidence has increased year by year (*Kelley et al., 2020*). It is the second most common primary liver malignancy after hepatocellular carcinoma (HCC) and it accounts for about 15% of all primary liver tumors and 3% of gastrointestinal cancers (*Bréchon et al., 2018*). Cholangiocarcinoma has a poor prognosis. Now some scholars use the ratio of neutrophil to eosinophil as an independent indicator to predict the prognosis of cholangiocarcinoma (*Sahin et al., 2024*), The Royal Marsden Hospital (RMH) Score has also been used to assess the prognosis of cholangiocarcinoma, and these studies have made great progress (*Sahin et al., 2024*).

Surgical treatment is the first choice for cholangiocarcinoma, but the onset of cholangiocarcinoma is latent, and most patients have lost the opportunity for surgery when diagnosed (*Chen et al., 2015*). It has been reported that the radical resection rate of intrahepatic cholangiocarcinoma is 30–40%, which is higher for other hepatobiliary malignancies (*El-Diwany, Pawlik & Ejaz, 2019*; *Wirth & Vogel, 2016*). As a result, its survival rate has been very low. The average postoperative tumor-free survival (DFS) currently ranging from 12 to 36 months worldwide (*Choi et al., 2009*; *Endo et al., 2008*). Recently, the immunotherapy of cholangiocarcinoma has also received attention, but the research methods and results are still not satisfactory (*Ricci, Rizzo & Brandi, 2020a*).

Adjuvant chemotherapy for advanced cholangiocarcinoma can degrade the staging, afford operation possibility (*Weigt & Malfertheiner, 2010*; *Kato et al., 2013*, *2015*). But the median survival was still less than 1 year (*Malka et al., 2014*; *Lee et al., 2012*; *Primrose et al., 2019*). In addition, chemotherapy also has many side effects, including anemia, neutropenia, thrombocytopenia, fatigue, fever, hypokalemia, and so on (*Li et al., 2016*; *Cai et al., 2020*). Targeted therapy is a way to treat malignant tumor by inhibiting apoptosis, proliferation, and activating or inhibiting specific cancer-related proteins (*Pérez-Herrero & Fernández-Medarde, 2015*). According to the study of *Ricci, Rizzo & Brandi (2020b)*, the occurrence and development of cholangiocarcinoma involves DNA damage repair (DDR) pathway, so the targeted therapy at the gene level has made rapid progress in recent years. Some studies have shown that targeted therapy can help patients achieve better survival time (*Harding et al., 2018*), but the benefits is limited. So there are many studies have combined targeted therapy with chemotherapy in the treatment of advanced cholangiocarcinoma (*Malka et al., 2014*; *Lee et al., 2012*; *Leone et al., 2016*; *Valle et al., 2015*). In these studies, the results of targeted therapy combined with chemotherapy were contradictory, with some studies showing superior (*Leone et al., 2016*; *Abou-Alfa et al., 2020*; *Demols et al., 2020*) and some studies showing inferior (*Malka et al., 2014*; *Santoro et al., 2015*; *Kim et al., 2020*; *Moehler et al., 2014*) results.

Therefore, this study for the first time conducted a network meta-analysis on the prognostic value of targeted therapy, chemotherapy and targeted combination chemotherapy in patients with advanced cholangiocarcinoma, so as to clarify the role of targeted therapy, chemotherapy and combination of them in the treatment of advanced

cholangiocarcinoma. We hope provide a high-quality evidence for clinical decision-making by our study.

## METHODS

We did a systematic review and network meta-analysis of placebo-controlled and head-to-head randomised controlled trials (RCT) according to PRISMA guidelines (*Hutton et al., 2015*). In addition, this study has been registered with PROSPERO, under the number CRD42022314255.

### Search strategy

We conducted a comprehensive search of databases including MEDLINE, Cochrane Central Register of Controlled Trials (CENTRAL), Embase, and PubMed from their inception to October 2023, with no language restrictions. The search strategy utilized terms related to targeted drugs for cholangiocarcinoma, alongside synonyms for advanced stages (such as late-stage, advanced, terminal), and Mesh terms pertaining to cholangiocarcinoma (including cholangiocarcinoma, bile duct neoplasms, bile duct cancers, cholangiocellular carcinomas). Additionally, we screened the reference lists of included studies and previous reviews for further relevant studies.

### Inclusion and exclusion criteria

We enrolled patients with advanced cholangiocarcinoma in our randomized controlled trial, regardless of their surgical history, provided they were currently ineligible for surgical treatment. Excluded from our study were patients with treatment resistance, prior chemotherapy, distant metastases, lack of histological confirmation, and an Eastern Cooperative Oncology Group (ECOG) performance status score of two or higher. Our study specifically focused on patients with inoperable advanced cholangiocarcinoma. We included all relevant targeted drugs for cholangiocarcinoma, such as Ivosidenib, Infigratinib, Pemigatinib, Panitumumab, Enasidenib. We excluded chemotherapy *vs.* chemotherapy studies. We included published RCT comparing targeted therapy with chemotherapy (GEMOX (*Razumilava & Gores, 2014*) or FOLFOX (*Eisenhauer et al., 2009*) scheme) or targeted therapy plus chemotherapy or placebo. We choose progression-free survival (PFS) and overall survival (OS) as to be the outcomes. According to RECIST v1.1 (*Eisenhauer et al., 2009*), Progression-Free Survival was assessed starting from the initiation of the first treatment until either the confirmation of disease progression or death. Overall Survival was calculated from the time of the first treatment until the occurrence of death or the last recorded follow-up. Excluded from our analysis were studies involving non-advanced cholangiocarcinoma, those lacking targeted drugs, conference reports, or reviews. Additionally, studies identified as having a high risk of bias in sequence generation or allocation concealment, as determined by the Cochrane Collaboration's risk of bias tool, were also omitted (*Cumpston et al., 2019*).

### Data extraction

Two reviewers independently screened the search results, retrieved full-text articles, and assessed them against inclusion criteria. Additionally, they extracted key information

including the first author's name, publication year, study design, country of origin, sample size, intervention and control details, and outcomes. Whenever there was uncertainty, a third reviewer provided input. Data extraction was conducted independently by two reviewers, and entered into electronic forms using Microsoft Excel. When standard errors were provided in the original study for both experimental and control groups, standard deviations were calculated using the formula: standard deviation = standard error × $\sqrt{n}$. In cases where standard errors were not available, standard deviations were estimated based on methods outlined in section 7.7.3 of the Cochrane Handbook for Systematic Reviews, utilizing confidence intervals, t-values, quartiles, ranges, or $p$-values. If necessary data were still unavailable, authors were contacted up to four times within a 6-week period to request the required information.

## Risk of bias

Risk of bias in RCTs for each individual study was assessed independently by L using the Cochrane Collaboration's risk of bias tool (*Cumpston et al., 2019*), which evaluated potential biases including selection bias (random sequence generation and allocation concealment), performance bias (blinding of patients and personnel), detection bias (blinding of outcome assessment), attrition bias (incomplete outcome data), reporting bias (selective outcome reporting), and other biases. Each study was categorized as having a low, high, or unclear risk of bias for each specific domain, with unclear assigned when reporting was insufficient to assess a particular bias domain.

## Data analysis

We conducted a network meta-analysis using both direct and indirect comparisons, utilizing STATA software (Version 14.0; Stata Corporation, College Station, Texas, USA) and R Software (Version 4.1.2). Effect sizes were measured as hazard ratios (HR) and mean differences (MD) for continuous efficacy-related outcomes and dichotomous outcomes respectively. Specifically, for PFS and OS, MD and HR values were extracted. These effect sizes were then synthesized using a random-effects network meta-analysis model. Summary MDs with 95% credible intervals (CrIs) and HRs with 95% CrIs for all pairwise comparisons were presented in a league table format, and the results of comparing outcomes between each intervention group and the placebo group were depicted using a forest plot. In the network plot, direct comparisons between two interventions were represented by connections, with node size reflecting the number of studies and edge thickness indicating the precision of the direct estimate for each pairwise comparison. We rank treatment effects according to $p$-scores (*Rücker & Schwarzer, 2015*). $p$-scores ranged from 0 to 1, a higher $p$-score indicating a greater degree of treatment effects. We use the $\tau^2$ test and $p$-value to qualitatively analyze the statistical heterogeneity between the studies. The larger the $\tau^2$ and the smaller the $p$-value, the greater the possibility of heterogeneity; on the contrary, the smaller the existence heterogeneity. In addition, $I^2$ is a parameter for quantitative analysis of heterogeneity among study results. It's values are distributed between 0 and 100%. When $I^2$ is less than 25%, the heterogeneity is low; the heterogeneity was moderate between 25% and 50%; $I^2 > 75\%$ indicates high heterogeneity. In summary,

when $I^2 > 50\%$, it indicates that there is a large heterogeneity. We will use both global and local approaches to check for inconsistencies in research results. For global inconsistency, we used the design-by-treatment test to statistically assess the inconsistencies (*Higgins et al., 2012*). In addition, we will assessment of local inconsistency by separating indirect from direct evidence using the R netmeta package (*Dias et al., 2010*).

## RESULTS

### Literature search and heterogeneity test

We are in the database retrieval to a total of 1,336 articles, and 23 full-text articles were retrieved. Two investigators confirmed the outcomes of interest by viewing the full text, and finally included 13 studies (*Bréchon et al., 2018*; *Chen et al., 2015*; *Malka et al., 2014*; *Lee et al., 2012*; *Primrose et al., 2019*; *Leone et al., 2016*; *Valle et al., 2015*; *Abou-Alfa et al., 2020*; *Demols et al., 2020*; *Santoro et al., 2015*; *Kim et al., 2020*; *Moehler et al., 2014*; *Vogel et al., 2018*) with 1,914 participants (Fig. 1) from 2014 to 2020. The average age was 62 and the youngest is 18 and the oldest is 83, males accounted for 50.1% of total. The characteristics of the included literature are shown in Table 1.

### Literature quality evaluation

We excluded studies with high risk of bias for randomisation and allocation, but methods for sequence generation and allocation concealment were often not described in detail and, therefore, were coded as unclear. The number of studies with high, unclear, and low risk of bias for the individual items was: 0, 7, and 6 for randomisation, 4, 5, and 4 for allocation concealment, 7, 2, and 4 for blinding of patients and personnel, 1, 8, and 4 for rater blinding, 0, 2, and 11 for missing outcomes, 2, 1, and 10 for selective reporting, and 0, 1, and 12 for other biases (Fig. 2).

### Network meta-analysis

#### Comparison of HR in OS

Among the included literature, 12 studies involving 1,743 patients with advanced cholangiocarcinoma provided HR values in OS for analysis. The tenth study provided HR values of OS of two different populations, which were separately included in this study (Fig. 3A). Compared with placebo (Fig. 4A), the TT+CT, chemotherapy (CT), and targeted therapeutic (TT) groups all significantly reduced HR in patients with advanced cholangiocarcinoma (see the results from forest plot). From the league table (Fig. 5A) we can know the TT+CT is significantly superior to TT group ($p < 0.05$), and TT+CT group has the best effect according to $p$-scores. This result does not form a closed loop, so there is no global inconsistency and point inconsistency.

#### Comparison of MD in OS

Of the included literature, 10 studies included 1,683 patients with advanced cholangiocarcinoma, providing Mean ± SD values in OS for analysis. Article 7 is a three-arm study, which is included in our study (Fig. 3B). Compared with placebo (Fig. 4B), the MD of patients with advanced cholangiocarcinoma was increased in
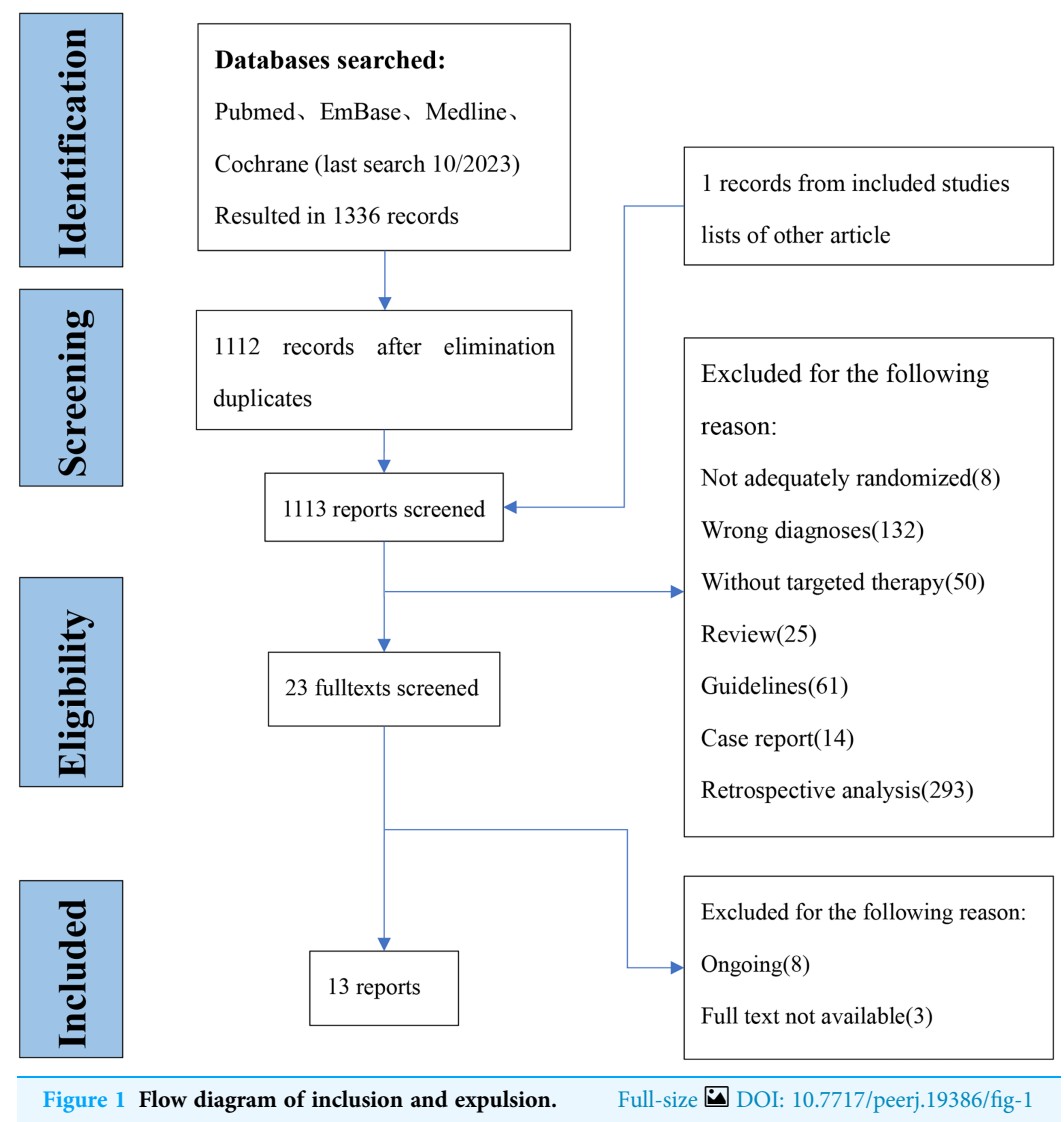

**Figure 1** Flow diagram of inclusion and expulsion.

TT+CT, CT and TT groups (see the results from forest plot). We can know from the league table (Fig. 5B), The TT+CT was superior to other groups, but there was no statistical significance ($p > 0.05$). According to $p$-scores, The TT+CT group had the best effects. There was no global inconsistency ($p > 0.05$) and no point inconsistency ($p > 0.05$).

### Comparison of HR in PFS

Twelve studies involving 1,467 patients with advanced cholangiocarcinoma provided HR values for PFS analysis and the seventh and tenth studies provided HR of PFS in two groups, which were respectively included in this study (Fig. 3C). HR of patients with advanced cholangiocarcinoma was decreased in TT+CT, CT and TT groups when Compared with placebo (Fig. 4C) (see the results from forest plot). TT+CT group is significantly superior to TT group ($p < 0.05$) which comes from the league table (Fig. 5C),

**Table 1 Characteristics of subjects in eligible studies.**

| Author | Year | Study design | Country | Sample size | Intervention | Control | Outcomes |
|---|---|---|---|---|---|---|---|
| Ghassan. K. Abou-Alfa (*Ricci, Rizzo & Brandi, 2020b*) | 2020 | RCT | USA | 187 | Ivosidenib | Placebo | OS: HR, Mean PFS: HR, Mean |
| A. Demols (*Harding et al., 2018*) | 2020 | RCT | Belgium | 66 | Regorafenib | Placebo | OS: HR, Mean PFS: HR, Mean |
| Francesco Leone (*Cai et al., 2020*) | 2016 | RCT | Italy | 89 | GEMOX+Panitumumab | GEMOX | OS: HR, Mean PFS: HR, Mean |
| David Malka (*Weigt & Malfertheiner, 2010*) | 2014 | RCT | France | 150 | GEMOX+Cetuximab | GEMOX | OS: HR, Mean PFS: HR, Mean |
| Jeeyun Lee (*Kato et al., 2013*) | 2012 | RCT | South Korea | 268 | GEMOX+Erlotinib | GEMOX | OS: HR, Mean PFS: HR, Mean |
| Juan. W. Valle (*Pérez-Herrero & Fernández-Medarde, 2015*) | 2015 | RCT | UK | 124 | Cisplatin+Gemcitabine +cediranib | Cisplatin +Gemcitabine +Placebo | OS: HR, Mean PFS: HR, Mean |
| A. Santoro (*Leone et al., 2016*) | 2014 | RCT | Italy | 173 | Vandetanib Vandetanib +Gemcitabine | Gemcitabine +Placebo | OS: Mean PFS: HR, Mean |
| Richard D. Kim (*Valle et al., 2015*) | 2020 | RCT | USA | 44 | Trametinib | 5-FU/Lv or Capecitabine | OS: HR, Mean PFS: HR |
| John N. Primrose (*Kato et al., 2015*) | 2019 | RCT | UK | 447 | Capecitabine | Placebo | OS: HR, Mean PFS: N |
| M. Moehler (*Abou-Alfa et al., 2020*) | 2014 | RCT | Germany | 97 | Gemcitabine+Sorafenib | Gemcitabine +Placebo | OS: HR, Mean PFS: HR, Mean |
| Arndt Vogel (*Cumpston et al., 2019*) | 2018 | RCT | Germany | 90 | Cisplatin+Gemcitabine +panitumumab | Cisplatin +Gemcitabine | OS: HR, Mean PFS: N |
| Marie Bréchon (*Bréchon et al., 2018*) | 2017 | RCT | France | 57 | GEMOX+bevacizumab | GEMOX | OS: HR, Mean PFS: HR, Mean |
| J. S. Chen (*Sahin et al., 2024*) | 2015 | RCT | Taiwan | 102 | GEMOX+cetuximab | GEMOX | OS: HR, Mean PFS: HR, Mean |

and TT+CT group has the best effect according to $p$-scores. This result does not form a closed loop, so there is no global inconsistency and point inconsistency.

### Comparison of MD in PFS

Ten studies included 1,333 patients with advanced cholangiocarcinoma, which provided us with mean values for PFS analysis. Among them, the seventh three-arm study was separately included (Fig. 3D). The TT+CT, CT and TT groups all increased MD in patients with advanced.

Cholangiocarcinoma when compared with placebo (Fig. 4D) (the results comes from forest plot), but there was statistical significance only in TT group ($p < 0.05$). We can know from the league table (Fig. 5D), the TT+CT was superior to other groups, but there was no statistical significance ($p > 0.05$). According to $p$-scores, TT+CT group had the best effect. There was no global inconsistency ($p > 0.05$) and no point inconsistency ($p > 0.05$).

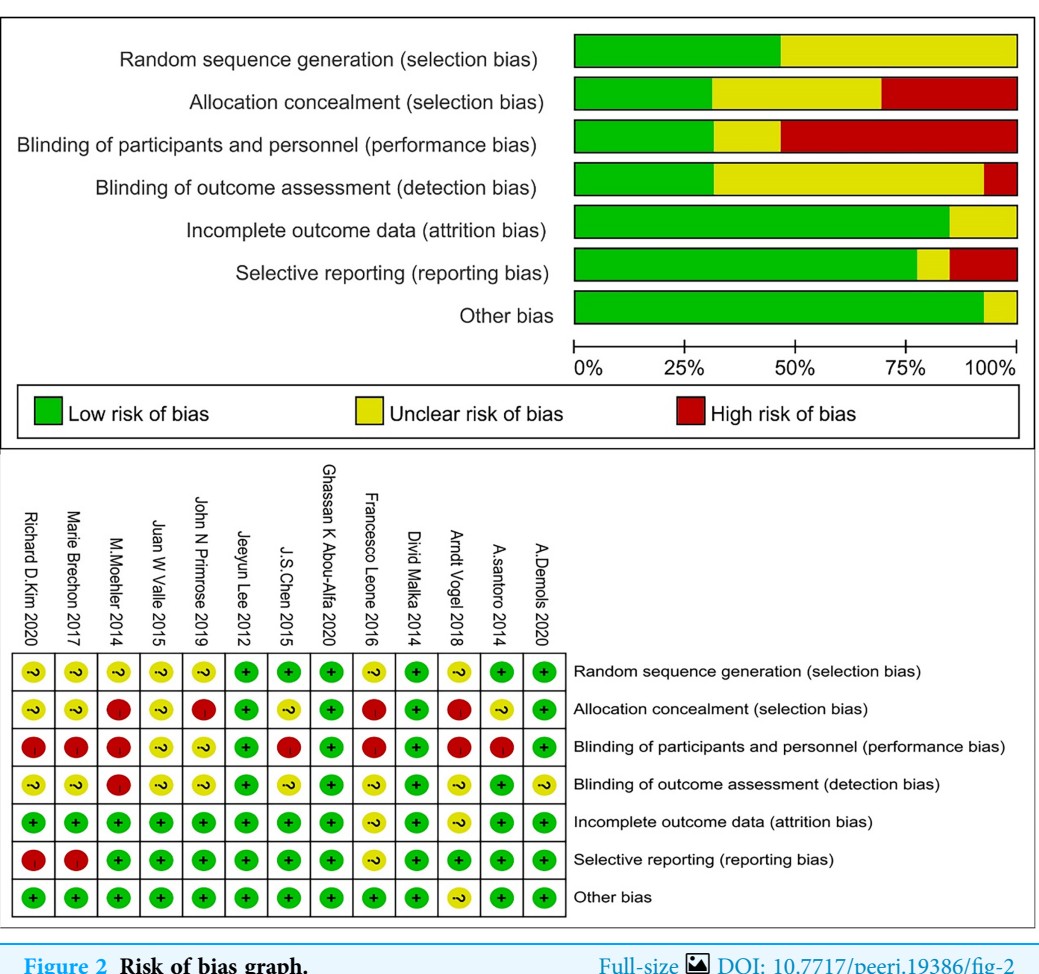

**Figure 2  Risk of bias graph.**               

## DISCUSSION

Our study is the first network meta-analysis of efficacy of targeted plus chemotherapy, targeted therapy alone and chemotherapy alone in advanced cholangiocarcinoma. The main findings were as follows: 1) All three treatments significantly reduced the hazard ratio in OS and PFS when compared with Placebo; 2) HR and MD values of OS and PFS in TT + CT group were significantly better than those in the other two groups; 3) Among the three therapeutic measures, only targeted therapy alone significantly increased the progression-free survival time of patients, thereby improving the quality of life of patients. Therefore, the prospect of targeted therapy is broad.

All three treatments compared with the placebo significantly reduced the hazard ratios in OS and PFS, consistent with previous studies (*Valle et al., 2010*).

Targeted therapy plays an antitumor role by combining drugs with related carcinogenic targets, such as epidermal growth factor receptor (EGFR), vascular epidermal growth factor (VEGF) and its receptor (VEGFR), *etc*. When targeted drugs combine with these targets, tumor proliferation and angiogenesis can be inhibited. Mutations in isocitrate dehydrogenase (IDH), for example, give cells the ability to produce hydroxy-glutaric acid

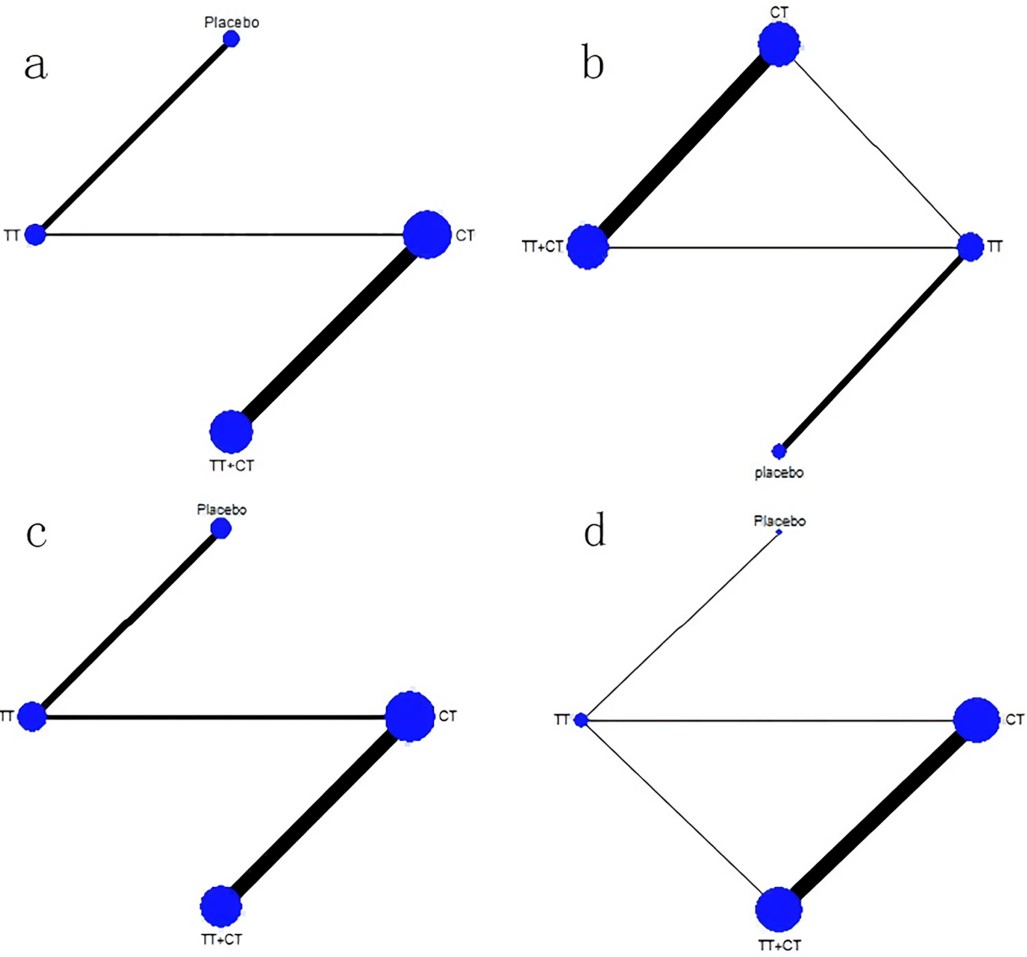

**Figure 3 Network plot.** (A) Network plot of HR that from OS. (B) Network plot of MD that from OS. (C) Network plot of HR that from PFS. (D) Network plot of MD that from PFS. Treatments with direct comparisons are linked with a line; its thickness corresponds to the number of trials evaluating the comparison. CT = chemotherapy. TT = targeted therapeutic. TT+CT = targeted therapy + chemotherapy.

(2-HG), whose continued production is important for the development of tumors (*Harding et al., 2018*). In the research of *Zhu et al. (2021)*, 187 patients were randomly assigned to receive ivosidenib ($n = 126$) or placebo ($n = 61$), median OS was 10.3 months (95% CI [7.8–12.4] months) with ivosidenib *vs.* 7.5 months (95% CI [4.8–11.1] months) with placebo (hazard ratio, 0.79 (95% CI [0.56–1.12]); 1-sided $p = 0.09$). When adjusted for crossover, median OS with placebo was 5.1 months (95% CI [3.8–7.6] months; hazard ratio, 0.49 (95% CI [0.34–0.70]); 1-sided $p < 0.001$). Unfortunately, this result was not statistically significant ($p > 0.05$).

In previous studies, TT+CT had better HR in OS and PFS than the other two groups (*Leone et al., 2016*; *Abou-Alfa et al., 2020*; *Demols et al., 2020*), but some were worse (*Malka et al., 2014*; *Santoro et al., 2015*; *Kim et al., 2020*; *Moehler et al., 2014*). In response to this controversy, this study combined their results, and the final result was that the

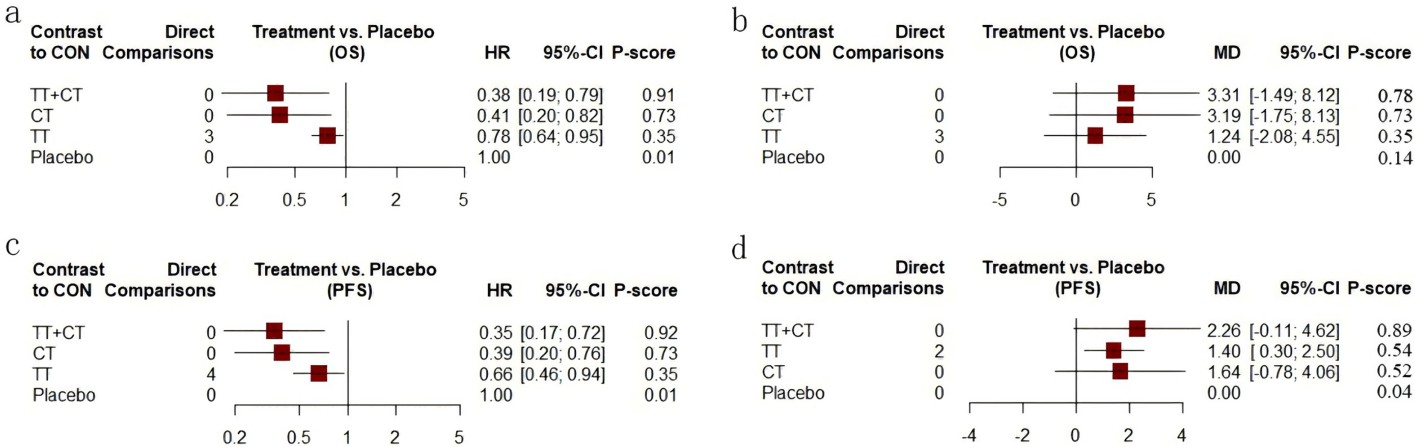

**Figure 4 Forest plot.** (A) Forest plot of HR that from OS. (B) Forest plot of MD that from OS. (C) Forest plot of HR that from PFS. (D) Forest plot of MD that from PFS. CT = chemotherapy. TT = targeted therapy. TT+CT = targeted therapy + chemotherapy.

**Figure 5 League table.** (A) League table of HR that from OS. (B) League table of MD that from OS. (C) League table of HR that from PFS. (D) League table of MD that from PFS. Treatments in diagonal cells are reported in order of relative ranking for efficacy. Results of the network meta-analysis are presented in the left lower half and results from pairwise comparisons in the upper right half. CT = chemotherapy. TT = targeted therapy. TT+CT = targeted therapy + chemotherapy.

TT+CT group had significantly better HR in OS and PFS than other two groups. This result may be attributed to the synergistic effect of combination therapy on targeted therapy and chemotherapy. Chemotherapy inhibits tumor cell proliferation in a wider range and reduces tumor recurrence and metastasis, while targeted therapy inhibits cancer targets more precisely, thus benefiting patients more. For example, in the study of *Lee et al. (2012)*, 133 patients were randomly assigned to the chemotherapy alone group and 135 to the chemotherapy plus erlotinib group. The median progression-free survival was 4.2 months (95% CI [2.7–5.7]) in the chemotherapy alone group and 5.8 months (95% CI [4.6–7.0]) in the chemotherapy plus erlotinib group (HR 0.80, 95% CI [0.61–1.03]; $p = 0.087$). It could be seen that PFS in the targeted combined chemotherapy group was larger, but this result was not statistically significant ($p > 0.05$). This may be related to the deficiency of this study: it was not clear whether there were deletion, mutation and rearrangement of corresponding genes in the included population. If the patient does not have the appropriate genetic changes, the targeted therapy is meaningless and the results of the study are inaccurate. Therefore, we hope that in the future, more medical centers will

include genetic testing as a routine test, so that we can conduct RCT specifically if we know the type of disease changes in patients. This kind of research will greatly facilitate the development and application of targeted drugs, provide high-quality evidence for disease management decisions, and bring more choices to patients.

Only targeted therapy alone significantly increased progression-free survival, which has not been shown in previous studies. *Gutman et al. (2013)* defined PFS as the time between the start of randomization and the progression of the tumor (in any way) or death (for any reason). *Gutman et al. (2013)* also reported that PFS significantly affected patients' quality of life and disease severity. This means that targeted drugs can significantly improve patients' quality of life and reduce disease symptoms for our study. It also indicated that PFS significantly affected patients' quality of life and disease severity. This conclusion may have something to do with the way chemotherapy drugs work: Chemotherapy works on all cells, including normal cells. Frequent side effects after chemotherapy include neutropenia, anemia, and thrombocytopenia (*Valle et al., 2010*), which leads to low immunity in patients and significantly increases the probability of immune escape of tumor cells (*Zhu et al., 2021*). After immune escape, tumor cells continue to proliferate and metastasize, eventually leading to disease progression. Therefore, there is less benefit from PFS in either chemotherapy alone or in targeted combination chemotherapy. But the hypothesis still needs to be confirmed by researchers. This conclusion is also likely to be inaccurate, because the literature included in this study is small and biased. We also hope that in the future there will be more treatments for advanced cholangiocarcinoma, such as immunotherapy combined targeting, which is very promising.

The shortcomings of this article are as follows: 1) The included studies are not all targeted therapies. Some studies do have strict criteria for inclusion. These studies accounted for 25% of all cases (480/1,914). But the rest of the research did not identify key information such as whether the patient's genes changed, what type of changes and where. Therefore, we are looking forward to the emergence of more targeted therapy RCTs that identify the type of genetic changes in the future. 2) Due to the small number of inclusion in this study, non-English articles were not included owing to quality concerns, which may lead to publication bias. Therefore, we hope that more RCTs with high quality, multi-center, multi-stratified and precise drug delivery can be conducted for advanced cholangiocarcinoma, so as to bring more evidence-based medical evidence for the diagnosis and treatment of patients with advanced cholangiocarcinoma and make contributions to changing the prognosis of patients with advanced cholangiocarcinoma.

### Funding

The authors received no funding for this work.

### Competing Interests

The authors declare that they have no competing interests.

## Author Contributions

- Zhoujun Liao conceived and designed the experiments, prepared figures and/or tables, and approved the final draft.
- Zhuoyue Yao conceived and designed the experiments, prepared figures and/or tables, and approved the final draft.
- Zhiqing Yang conceived and designed the experiments, prepared figures and/or tables, and approved the final draft.
- Shaohua Yang performed the experiments, prepared figures and/or tables, and approved the final draft.
- Wenjuan Gu conceived and designed the experiments, prepared figures and/or tables, and approved the final draft.
- Huijie Wang performed the experiments, prepared figures and/or tables, authored or reviewed drafts of the article, and approved the final draft.
- Lingyan Deng performed the experiments, authored or reviewed drafts of the article, and approved the final draft.

## Data Availability

This is a systematic review/meta-analysis.

## Supplemental Information

Supplemental information for this article can be found online at http://dx.doi.org/10.7717/peerj.19386#supplemental-information.

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
