# Peer review of "Comparative efficacy of targeted therapy, chemotherapy and their combination for advanced cholangiocarcinoma: a systematic review and network meta-analysis"

_PeerJ, doi:10.7717/peerj.19386_

## Round 0.1 · original submission · Major Revisions

Your work presents valuable findings as the first network meta-analysis in this area, and the reviewers have found it to be well-conducted and worthwhile. However, some revisions are needed to enhance its impact and clarity.
The manuscript would benefit from an expanded introduction that better reflects the current treatment landscape for biliary tumors. We recommend incorporating additional recent literature (particularly from 2022-2024) to provide a more comprehensive context. The discussion section needs elaboration to include future perspectives, knowledge gaps, and stronger clinical implications. We encourage you to share more expert interpretation of your findings and thoughts on how this field may develop over the next five years.
Your limitations analysis requires more depth, particularly regarding the retrospective nature of included studies, generalizability of findings, and potential selection bias. Some technical improvements are also needed, including defining abbreviations at first use (especially TT and CT), standardizing terminology, and addressing minor grammatical issues throughout the text.

Reviewer 1 ·

Basic reporting

no comment

Experimental design

no comment

Validity of the findings

no comment

Additional comments

The manuscript is well-written with clear and professional language. No significant grammatical or syntactical issues were noted. However, I suggest these edits to improve it more:
The Limitations section can be elaborated more. Specifically, consider addressing the retrospective nature of the study in more detail, the generalizability of the findings to other populations, and the potential for selection bias due to the exclusion criteria.

Reviewer 2 ·

Basic reporting

.

Experimental design

.

Validity of the findings

.

Additional comments

Dear Editor, thank you so much for inviting me to revise this manuscript about biliary tumors.

This study addresses a current topic.

The manuscript is quite well written and organized. English could be improved.
Figures and tables are comprehensive and clear.
The introduction explains in a clear and coherent manner the background of this study.

We suggest the following modifications:
• Introduction section: although the authors correctly included important papers in this setting, we believe the evolving systemic treatment scenario for biliary tumors should be further discussed and some recent papers added within the introduction, only for a matter of consistency. We think it might be useful to introduce the topic of this interesting study.
• Methods and Statistical Analysis: nothing to add.
• Discussion section: Very interesting and timely discussion. Of note, the authors should expand the Discussion section, including a more personal perspective to reflect on. For example, they could answer the following questions – in order to facilitate the understanding of this complex topic to readers: what potential does this study hold? What are the knowledge gaps and how do researchers tackle them? How do you see this area unfolding in the next 5 years? We think it would be extremely interesting for the readers.

However, we think the authors should be acknowledged for their work. In fact, they correctly addressed an important topic, the methods sound good and their discussion is well balanced.

One additional little flaw: the authors could better explain the limitations of their work, in the last part of the Discussion.

We believe this article is suitable for publication in the journal although major revisions are needed. The main strengths of this paper are that it addresses an interesting and very timely question and provides a clear answer, with some limitations.

We suggest a linguistic revision and the addition of some references for a matter of consistency. Moreover, the authors should better clarify some points.

Reviewer 3 ·

Basic reporting

The manuscript is generally well-written and presents novel findings as the first network meta-analysis examining targeted therapy plus chemotherapy, targeted therapy alone, and chemotherapy alone in advanced cholangiocarcinoma. I recommend this manuscript be accepted after minor revisions.

Experimental design

The abstract lacks definitions for abbreviations when first used (e.g., TT, CT). Please provide full terms at first mention.
On page 3, line 13, the word "insidious" does not fit well in this context. Please consider using more appropriate medical terminology to describe the disease characteristics.
On page 4, line 10, the term "early chemotherapy" is unclear. This needs to be clearly defined or rephrased for better clarity.

Validity of the findings

On page 5, line 43, please provide the full terms for TT and CT (Targeted Therapy and Chemotherapy) at their first appearance.

Additional comments

There are some minor spacing issues throughout the document that should be corrected during proofreading.

---

## Round 0.2 · accepted · Accept

Since authors have fully made revisions, I think this paper can be accepted for publication.

Reviewer 2 ·

Basic reporting

acceptance.

Experimental design

acceptance.

Validity of the findings

acceptance.

Additional comments

acceptance.

Reviewer 3 ·

Basic reporting

The manuscript is generally well-written and presents novel findings as the first network meta-analysis examining targeted therapy plus chemotherapy, targeted therapy alone, and chemotherapy alone in advanced cholangiocarcinoma. I recommend this manuscript be accepted.

Experimental design

The manuscript is generally meet the standards.

Validity of the findings

Fine

Additional comments

None